# Nurse and Patient Assessments of COVID-19 Care Quality in China: A Comparative Survey Study

**DOI:** 10.3390/ijerph20032717

**Published:** 2023-02-03

**Authors:** Wenjing Jiang, Jia Jiang, Xing’e Zhao, Zina Liu, Maritta A. Valimaki, Xianhong Li

**Affiliations:** 1Department of Nursing, Zigong First People’s Hospital, Zigong 643000, China; 2Xiangya School of Nursing, Central South University, Changsha 410013, China; 3West China School of Pharmacy, Sichuan University, Chengdu 610041, China; 4Department of Liver Transplantation, The Second Xiangya Hospital of Central South University, Changsha 410011, China

**Keywords:** caring behavior, comparison, COVID-19, nurses, patients, quality of care

## Abstract

During the COVID-19 pandemic, the quality of nursing care was a concern due to nurses’ overwhelming workload. A cross-sectional design was conducted to compare perceptions between nurses and patients about the quality of nursing care for COVID-19 patients and to explore factors associated with these perceptions. Data were collected during the COVID-19 pandemic from 17 March to 13 April 2020 in five hospitals in Wuhan, China. Perceptions of care quality were assessed among nurses and patients using the Caring Behaviors Inventory. Nurses rated the quality of caring behaviors higher than patients. Both nurses and patients rated technical caring behaviors at high levels and rated the item related to “spending time with the patient” the lowest, while patients rated it much lower than nurses. Nurses’ sex, participation in ethical training organized by the hospital, professional title, being invited to Wuhan, and length of working experience in years were significantly associated with nurses’ self-evaluated caring behaviors. Moreover, inpatient setting and communication mode were significantly associated with patients’ self-evaluated caring behaviors.

## 1. Introduction

Coronavirus disease 2019 (COVID-19) has been characterized as a pandemic by the World Health Organization [1]. Since its emergence in 2019, it has become the most severe public health crisis in living memory [2]. China faced the epidemic early [3], and the rapid increase in cases during the early stages placed a heavy burden on the medical system. Between 31 December 2019 and 2 February 2020, 79,394 COVID-19 cases were identified in China, along with 2838 deaths [4]. The high rates of deaths in the short term and the lack of effective medications threatened the public and exacerbated high psychological stress. Anxiety, irritability, loneliness, and helplessness were prevalent among patients [5,6]. Therefore, high-quality nursing care was urgently needed to support patients’ physical and psychological recovery [5,6,7]. 

COVID-19 is a highly contagious respiratory disease that requires a high level of occupational precaution. Nurses who worked in isolation wards and provided direct care for COVID-19 patients were equipped with standardized personal protective equipment (PPE) against infection, including protective suits, masks, gloves, goggles, and face shields [8]. Moreover, the protective equipment impacted nurses’ visibility and limited their communication with patients [9]. Heavy PPE, such as multilayer masks and gloves, increased the physical burden on nurses and caused inconvenience in performing nursing procedures, which might have further affected the quality of care [9]. 

Typically, self-evaluation measurements are used to evaluate the quality of caring behaviors among nurses [10,11,12]. Studies have revealed that nurses usually report a high level of caring behaviors [10,11,12]. However, the ability of self-administrative tools to measure nurses’ perceptions of care quality leading to socially desirable results has been doubted [13,14]. Another approach to evaluating quality in nursing care is describing patients’ perceptions toward nursing care [15,16,17]. This approach is important as patients’ perceptions of care quality could ultimately impact clinical outcomes and nurse–patient relationships; therefore, it provided more critical input to nursing practice reform [17,18]. Factors such as the presence of family caregivers [15] and person-centered care [16] affected patients’ perceptions about nurses’ caring behaviors. However, the above results were devoid of comparative evaluations from nurses’ and patients’ perspectives, and they could not truly reflect the consistency of the two parties’ perceived caring behaviors; thus, the relevant intervention measures that are significant for improving caring behaviors based on this were limited [19].

Since caring develops in an interpersonal relationship that affects both patients and nurses, consistency between nurse and patient perceptions of caring quality is expected [20]. Therefore, another approach to assess the quality of nursing care by comparing nurse and patient perceptions [21,22] allows the critical monitoring of the quality of care and the evaluation of the effectiveness of the care. One example of a measurement to simultaneously evaluate caring behaviors among nurses and patients is the Caring Behaviors Inventory (CBI). The CBI was developed by Wolf et al. [23,24] and adapted into a short version by Wu et al. [14], which was used to measure caring behaviors from both the perspectives of nurses and patients. Several comparative studies on nurse and patient perceptions of the quality of nursing care have been conducted in the United States [25], Turkey [26,27], the United Kingdom [28], Greece [21], China [19], and Indonesia [29]. However, the results have been inconsistent. Some studies have indicated that nurses rated caring behaviors at a comparatively higher level than patients [19,25,26,28]. One study showed nurses and patients rated caring behaviors at the same level [29], while another indicated that nurses rated caring behaviors at a comparatively lower level than patients [21]. 

However, as far as we know, only a few studies have compared the perceptions of the quality of caring behaviors among nurses and patients in public health emergencies, including the COVID-19 pandemic. The topic is important as new knowledge could be used to contribute to the improvement of care quality in similar clinical settings worldwide in the current era. Therefore, the present study aimed to (a) describe the quality of caring behaviors evaluated by both nurses and patients, (b) compare the consistency between nurse- and patient-evaluated caring behaviors, and (c) explore the associated factors influencing caring behaviors during the COVID-19 pandemic in Wuhan, China.

## 2. Materials and Methods

### 2.1. Design 

A cross-sectional study was conducted through an online survey of nurses and patients from 17 March to 13 April 2020 in Wuhan, China. 

### 2.2. Setting

The study was conducted at five hospitals in Wuhan, including three “designated hospitals” (which were urgently renovated by the Chinese government to receive COVID-19 patients with severe symptoms) and two “mobile cabin hospitals” (which were temporarily converted from exhibition halls and stadiums to receive COVID-19 patients with mild symptoms). The hospitals included in the study were chosen based on convenience.

### 2.3. Study Population 

A convenience sample of nurses and patients was invited to participate in the study. Both the nurses and patients were recruited from the same departments in the same five hospitals by contact persons (research nurses) appointed in advance at each department by the researcher. 

The inclusion criteria for nurses were as follows: (a) registered nurses, (b) working in designated hospitals for at least one week, (c) providing direct care for COVID-19 patients, and (d) being willing to participate in the study. Nurses who were confirmed as having a COVID-19 infection were excluded. A total of 247 questionnaires were returned from the nurses, and 235 with completed information were used in this study.

Patients were also recruited in terms of proximity. The inclusion criteria for patients were as follows: (a) confirmed with COVID-19 infection, (b) hospitalized for at least 1 week in the current hospital (to receive nursing care and be in a position to judge it), (c) at least 18 years of age, (d) able to speak and read in Chinese, and (e) willing to participate in the study. Suspected patients and severely ill patients on respiratory machines were excluded from the study. A total of 131 questionnaires were returned from the patients, and 126 with completed information were analyzed in this study.

### 2.4. Instruments

#### 2.4.1. Background Information Sheet

According to previous studies [8,17,19], the potential influencing factors identified for nurses’ assessment of care quality included age, sex, marital status, educational background, being the only child in the family, having children, having ethical training experience (on-the-job training on nursing ethics (over 2 h) organized by the hospital), professional title, the length of working experience in years, monthly income, employment type, being invited from outside Wuhan to work in hospitals, work setting, and days worked in isolation wards. Patients’ background information included age, sex, marital status, residence, educational background, work status, monthly income, inpatient setting, inpatient days in isolation wards, communication mode, and previous hospitalization experience.

#### 2.4.2. The Caring Behaviors Inventory 

The Chinese version of the Caring Behaviors Inventory (CBI-24) [30] was used to evaluate the quality of caring behaviors among nurses and patients. It was developed and revised by Wolf et al. [23,24] and Wu et al. [14]. The CBI-24 consists of 24 items that include four dimensions: assurance of human presence (eight items), professional knowledge and skills (five items), patient respectfulness (six items), and positive connectedness (five items) [30]. The first two dimensions reflect technical caring behaviors, and the latter two reflect expressive caring behaviors. All items use a 6-point rating scale (1 = never, 6 = always). The total score of all items is divided by 24 with an average range of scores between 1 and 6. The total score of all items in the subscales was divided by the number of the items and the subscale scores. Higher scores on the subscale and total scale represent a higher quality of caring behaviors. The CBI-24 has adequate internal consistency reliability when used with a Chinese population, with a Cronbach alpha of 0.96 for nurses and patients [30]. In this study, Cronbach’s alpha was 0.97 for nurses and 0.96 for patients, and it ranged from 0.86 to 0.93 for the four subscales. 

### 2.5. Data Collectionh

Data were collected from 17 March to 13 April 2020, during the pandemic in Wuhan. Given the infectious nature of COVID-19 and the need for personal protection, the questionnaire was administered through an online professional survey platform (http://www.sojump.com, accessed on 17 March 2020), consisting of the instruments described above. Contact persons were responsible for recruiting the targeted nurses and patients in each hospital. Specifically, a flyer with a QR code linking to the screening sheet and questionnaires was sent to the targeted nurses through a WeChat group (an instant online chatting platform) in each hospital by the contact person. In addition, the contact persons directly recruited the targeted patients at the bedside using a similar flyer. Those who were interested in this study could scan the QR code to access the questionnaire via a smartphone or computer. Participants were required to read the informed consent at first. If they agreed to participate in the study, they could click “agree” to start filling out the following survey. Log-in IDs were recorded automatically, and thus, the person connected with each smartphone or computer ID could only submit the survey once. Participants could withdraw at any time. No personal identification numbers were collected, and all the information was stored by the Sojump online company, which had a confidentiality contract with the research team. 

### 2.6. Ethical Considerations

The study was approved by the Institutional Review Board of Xiangya Nursing School in Central South University (approval no. E202023). Target participants were given electronic informed consent explaining the aim of the study and assuring them of the anonymity of the collected data. They were also advised that they could refuse participation or withdraw from the study at any time without any consequences. Furthermore, their obligations were made clear about the potential benefits and risks of participating in this study. The collected data were protected by restricted access. Data were only used for this study and destroyed at the end of the study.

### 2.7. Data Analysis

Data analysis was performed using SPSS 18.0. (SPSS Inc., Chicago, IL, USA )Categorical variables were presented as frequencies and percentages. Continuous variables were presented as means and standard deviations. A *t*-test or one-way analysis of variance (ANOVA) was adopted to conduct the univariate analysis of normal distribution data, followed by post hoc tests. Multiple linear regression analysis using a stepwise regression model (enter *p* = 0.05, exclude *p* = 0.10) was conducted to determine the associated factors for caring behaviors. A *p*-value (two-sided) of less than 0.05 was considered significant.

## 3. Results

### 3.1. Participants’ General Characteristics

In all, 235 nurses were enrolled, and their general characteristics are summarized in Table 1. The nurses’ average age was 31.18 ± 5.05 years (range: 20–52 years), and 81.7% were female. In addition, 71.9% reported having ethical training experiences. Senior nurses accounted for half of the participating nurses. Moreover, 46.8% worked for 6–10 years, and 90.6% were dispatched to Wuhan.

In all, 126 patients were enrolled, and their general characteristics are summarized in Table 2. The patients’ average age was 52 years (52.63 ± 14.18), and 61.9% were female. The majority of patients (94.4%) reported communicating with nurses face to face, and 18.3% had previous hospitalization experiences.

### 3.2. Comparison of Nurse and Patient Self-Evaluated Caring Behaviors

Overall, both nurses and patients evaluated nursing care quality as high, with a total mean CBI-24 score of 5.32 ± 0.72 for nurses and 5.14 ± 0.70 for patients, although the differences were significant (*p* < 0.05). Nurses rated the quality of caring behaviors higher than patients, especially in expressive caring behaviors, namely, patient respectfulness (5.30 vs. 5.01, *p* < 0.01) and positive connectedness (5.17 vs. 4.87, *p* < 0.05). In addition, both groups rated technical caring behaviors, namely, professional knowledge and skills (5.39 vs. 5.35, *p* > 0.05) and the assurance of human presence (5.38 vs. 5.27, *p* > 0.05), at high levels (Table 3). Regarding the CBI-24 items, spending time with the patient was the lowest in both nurse and patient data (Table 4).

### 3.3. Factors Affecting Nurse and Patient Self-Evaluated Caring Behaviors

The total score for CBI-24 rated by nurses was significantly associated with sex, marital status, educational background, ethical training experience, professional title, the length of working experience in years, and whether being dispatched to Wuhan (all *p* < 0.05). Inpatient setting and communication mode (all *p* < 0.05) exhibited significant associations with CBI-24 total score rated by patients (Table 1 and Table 2).

Multiple linear regression models showed that nurses who were female (*β* = 0.266, *p* < 0.05), who had ethical training experience (*β* = 0.286, *p* < 0.01), and who were dispatched to Wuhan (*β* = 0.384, *p* < 0.01) had higher total scores for CBI-24. Interestingly, compared to junior nurses, senior nurses self-evaluated a relatively lower quality of caring behaviors (*β* = −0.256, *p* < 0.001), and similarly, compared to nurses with shorter working experience, in years, nurses with longer working experience self-evaluated a relatively lower quality of caring behaviors (*β* = −0.115, *p* < 0.05). 

In addition, patients who lived in designated isolation wards (compared to those living in mobile cabin isolation wards, *β* = 0.516, *p* < 0.01) and had face-to-face communication with nurses (compared to other communication modes, *β* = 0.661, *p* < 0.05) rated a higher total score of CBI-24 (Table 5).

## 4. Discussion

To the best of our knowledge, ours is one of the few studies to explore the quality of nursing care from the perspectives of nurses and patients during the COVID-19 pandemic. Our findings revealed that both nurses and patients reported high-quality caring behaviors, except for spending time with patients. Moreover, there was some inconsistency between nurses’ and patients’ self-evaluated caring behaviors, which provided evidence and suggestions on how to improve the quality of care for COVID-19 patients in similar clinical settings globally. 

### 4.1. Congruence between Nurse and Patient Self-Evaluated Caring Behaviors

In this study, the overall score and the scores obtained for each subscale of the CBI for both nurses and patients were very high (the average mean item score was over 5 for every item with the potential range of 1–6), which was contrary to the assumption that the pandemic might affect the quality of caring behaviors due to the fear of becoming infected in the pandemic [9]. Moreover, the scores of caring behaviors in our study were much higher than those reported in other studies, both from the perspectives of nurses and patients [19,21,25,28,31]. This inconsistency may be due to the unprecedented humanistic care policies implemented during the pandemic, including exerting extraordinary efforts for the treatment of patients [32], preparing daily necessities and food for patients [7], establishing a scientific and reasonable shift schedule [8], and providing a large amount of high-quality PPE to nurses. As a result, nurses demonstrated a high level of professional commitment and compassion during the outbreak, and patients were satisfied with the help and receiving social attention, which has been described elsewhere [33]. 

Categories that were scored higher by both nurses and patients included professional knowledge and skills and assurance of human presence, which meant nurses and patients all agreed that nurses could help patients meet technical needs more frequently than expressive needs. This is not a new phenomenon that emerged during this global pandemic since several previous studies have reported similar results [19,25,28,31]. Fundamentally, Chinese nursing education (especially in vocational schools) emphasizes basic nursing skills training, and not all schools make a psychological nursing course compulsory [19], which may lead to nurses practicing better technical caring skills than expressive caring skills. In most Chinese hospitals, psychological care is seldom evaluated in annual performance appraisals by hospital managers, which results in nurses paying less attention to patients’ emotional needs [19]. Furthermore, our findings also agree with previous studies, which have indicated that nurse shortages, multiple tasks, and complex working conditions resulted in less time for nurses to deal with patients’ emotional reactions [19,34].

An alarming finding was that the item related to spending time with the patient was rated as the lowest in terms of scores. On the one hand, during the early stage of the COVID-19 pandemic, partly because of the scarcity of protective resources and the fear of infection [35,36], nurses might consciously spend less face-to-face contact time with patients. On the other hand, COVID-19 patients might require more accompanying time from nurses than other patients due to isolation from their family members [5,6]. As a result, patients also rated this item with a relatively low score. Moreover, patients rated this item much lower than nurses. This gap between nurse and patient evaluations also deserves further attention.

### 4.2. Inconsistency between Nurse and Patient Self-Evaluated Caring Behaviors

Although nurses and patients both provided high scores for caring behaviors, nurses’ scores were higher than those of patients, which is consistent with previous studies [19,25,28]. First, from the perspectives of the provider and the recipients of caring behaviors, the tendency for this evaluation is justifiable. For example, nurses might be affected by professional and social desirability tendencies; thus, their self-evaluated care quality tends to be high [13,14]. Patients being care recipients means that their evaluations may be more rigorous [17,18]. This result raises concerns about the sensitivity of nurses in understanding and responding to patients’ actual and perceived needs and expectations. The result also reinforces the importance of enhancing the capacity to provide good care quality among nurses from educational and managerial perspectives by nurse educators and head nurses, respectively, as well as by the nursing community in general.

Specifically, patients scored lower than nurses in patient respectfulness and positive connectedness, which was consistent with several previous reports [19,21,31]. This means that nurses’ expressive caring behaviors might not be congruent with patients’ expectations or needs. This inconsistency between nurses and patients may be due to two reasons. First, the unexpected infection of COVID-19 seriously affected the patient’s psychological health [5,6], and they needed much more emotional support [34], such as having someone to talk with or listen to them, not necessarily to complain but to express their current emotions [37]. Nurses were the main persons whom patients could communicate with in the isolated hospitals; however, due to the heavy protective masks, patients may have been unsatisfied with this kind of communication approach. In addition, internalized stigma against patients due to the infectious nature of COVID-19 was quite high [5], which also made them more sensitive to disrespectful behaviors. In this study, we found statistically significant differences in nurses’ and patients’ responses. However, the differences were small. It is, therefore, difficult to interpret the clinical meaning of these results. In order to ensure the clinical impact of these results, a rigorous study with a larger sample size is needed.

### 4.3. Factors Influencing Nurse and Patient Self-Evaluated Caring Behaviors

Our study results indicated that female nurses and those with ethical training provided higher scores for caring behaviors, which was consistent with previous studies [20,38]. Nurses who were invited to Wuhan self-evaluated higher quality caring behaviors because they all participated in Wuhan’s anti-pandemic work voluntarily and, therefore, demonstrated more professional commitment and compassion [39], which might have led to higher quality caring behaviors. Interestingly, senior nurses and experienced nurses self-evaluated a relatively lower quality of caring behaviors, which was inconsistent with a previous study [40]. Future qualitative studies are needed to understand the reason for this. 

In addition, the present study indicated that patients living in designated hospital isolation wards provided higher caring scores than those living in the mobile cabin hospital isolation wards. This might be explained by the lower demand for caring behaviors by patients in the mobile cabin hospital isolation wards due to the relatively mild condition and the relatively low quota of nursing staff, which may have led to relatively low caring attention to each patient. Patients in the designated hospital had more severe conditions and were more psychologically stressed; however, the quota of nursing staff was sufficient, and the patients received more attention; thus, they were in a position to evaluate a higher level of caring quality. Our results also implied that patients who communicated with nurses face to face rated higher caring scores than those who communicated with nurses through indirect methods, such as bedside pagers and mobile phones. Face-to-face communication is an important approach for nurses to provide expressive caring [41], a crucial determinant of caring behaviors. Face-to-face communication also contributes to establishing a trustful relationship and encouraging patients to participate in decision making regarding their care [41].

### 4.4. Limitations of the Study

This study had several limitations. First, the study used a convenience sample of nurses (without non-registered nurses) and patients from five hospitals in Wuhan, China, which might not represent all anti-pandemic nurses and patients in China. Second, the online survey data collection method has its own limitations on data reliability. Nevertheless, when necessary, the contact persons guided patients to fill out the survey at bedside. Moreover, we set detailed instructions for filling out the survey, and we restricted the IP address to avoid repeatedly filling out the survey. Third, although the assessment of perceptions of caring behaviors through measurement tools provides quantifiable data suitable for comparisons, there is still a need to explore caring behaviors from a qualitative perspective. A combination of qualitative and quantitative data could better highlight the complexity of caring behaviors for COVID-19 patients. Fourth, we did not assess patients’ fear or which hospital our participants came from in this study, which might be influencing factors. 

## 5. Conclusions

The study results clearly demonstrated that both nurses and patients perceived nurse caring behavior as high quality during the COVID-19 pandemic. However, some inconsistency was still found, especially in the time nurses spent with the patient. Moreover, nurses reported higher-quality expressive caring behaviors than patients did. As nurses’ evaluations varied based on their socioeconomic background, deeper knowledge is needed to understand this rationale using qualitative methods. In addition, how to improve effective communication between nurses and patients in isolation rooms should also be explored in future studies. 

## Figures and Tables

**Table 1 ijerph-20-02717-t001:** Association between general information variables and nurses’ self-evaluated caring behaviors (*n* = 235).

Variables	*N* (%)	Caring Behaviors		
		Mean (*SD*)	*p*	Post Hoc
Age				
≤30	119 (50.6)	5.34 (0.76)	0.709	
>30	116 (49.4)	5.30 (0.67)		
Sex				
Male	43 (18.3)	5.02 (1.06)	0.037	
Female	192 (81.7)	5.38 (0.60)		
Marital status				
Single ^1^	61 (26.0)	5.36 (0.63)	0.008	1, 2 < 3
Married ^2^	155 (66.0)	5.36 (0.65)		
Divorced/widowed ^3^	19 (8.0)	4.83 (1.20)		
Educational background				
Certificate (technical school) ^1^	6 (2.6)	4.28 (1.36)	<0.001	2, 3 > 1 3 > 4
Associate degree ^2^	33 (14.0)	5.24 (0.80)		
Bachelor’s degree ^3^	180 (76.6)	5.41 (0.62)		
Master’s degree ^4^	16 (6.8)	4.89 (0.88)		
Whether being the only child in the family				
Yes	104 (44.3)	5.29 (0.75)	0.535	
No	131 (55.7)	5.34 (0.69)		
Children				
Yes	146 (62.1)	5.35 (0.67)	0.344	
No	89 (37.9)	5.26 (0.79)		
Ethical training experience				
Yes	169 (71.9)	5.39 (0.68)	0.009	
No	66 (28.1)	5.12 (0.77)		
Professional title ^a^				
Primary nurse ^1^	23 (9.8)	5.37 (0.58)	<0.001	1, 2, 3 > 4 > 5
Senior nurse ^2^	131 (55.7)	5.44 (0.57)		
Supervisor nurse ^3^	65 (27.7)	5.31 (0.72)		
Deputy chief nurse ^4^	12 (5.1)	4.52 (1.10)		
Chief nurse ^5^	4 (1.7)	3.43 (0.29)		
The length of working experience in years				
≤5 ^1^	29 (12.3)	5.51 (0.44)	<0.001	1, 2 > 4 > 5 3 > 5
6–10 ^2^	110 (46.8)	5.43 (0.61)		
11–15 ^3^	55 (23.4)	5.35 (0.69)		
16–20 ^4^	22 (9.4)	5.08 (0.85)		
≥21 ^5^	19 (8.1)	4.54 (1.00)		
Monthly income (RMB)				
≤3999 (≤USD 612.8)	15 (6.4)	5.12 (0.94)	0.574	
4000–5999 (USD 612.8–USD 919.2)	82 (34.9)	5.31 (0.71)		
6000–7999 (USD 919.2–USD 1225.6)	76 (32.3)	5.4 (0.64)		
8000–9999 (USD 1225.6–USD 1532)	41 (17.4)	5.33 (0.78)		
≥10,000 (≥USD 1532)	21 (8.9)	5.17 (0.72)		
Employment type				
State-employed	77 (32.8)	5.24 (0.77)	0.252	
Contract-employed	158 (67.3)	5.36 (0.69)		
Whether being dispatched to Wuhan				
Yes	213 (90.6)	5.35 (0.71)	0.048	
No	22 (9.4)	5.03 (0.76)		
Working setting				
Designated hospital intensive care unit	74 (31.5)	5.29 (0.84)	0.607	
Designated hospital isolation ward	126 (53.6)	5.36 (0.60)		
Mobile cabin hospital isolation ward	35 (14.9)	5.23 (0.83)		
Days worked in isolation wards				
≤30	119 (50.6)	5.32 (0.72)	0.975	
>30	116 (49.4)	5.32 (0.71)		

Note: ^a^—in China, primary nurses (Level 1) are new nurses who have recently graduated and passed the National Nurse Practitioner Registration Examination; senior nurses (Level 2) are nurses who have had 1–5 years of experience after Level 1 and who have passed the primary title examination; supervisor nurses (Level 3) are nurses who have had 1–7 years of experience after Level 2 and have passed the intermediate title examination; deputy chief nurse (Level 4) are nurses who have had 2–7 years of experience after Level 3 and have passed senior title examination; chief nurses (Level 5) are nurses who have had 5 years of experience after Level 4 and have specified academic performance.

**Table 2 ijerph-20-02717-t002:** Association between general information variables and patients’ self-evaluated caring behaviors (*n* = 126).

Variables	*N* (%)	Caring Behaviors	
		Mean (*SD*)	*p*
Age			
≤55	63 (50.0)	5.07 (0.76)	0.242
>55	63 (50.0)	5.21 (0.63)	
Sex			
Male	48 (38.1)	5.16 (0.7)	0.769
Female	78 (61.9)	5.12 (0.71)	
Marital status			
Single	12 (9.5)	5.39 (0.54)	0.203
Married/divorced/widowed	114 (90.5)	5.11 (0.71)	
Residence			
Urban area	119 (94.4)	5.14 (0.71)	0.703
Rural area	7 (5.6)	5.04 (0.55)	
Educational background			
Middle school or lower	30 (23.8)	5.11 (0.66)	0.236
High school	46 (36.5)	5.05 (0.74)	
Associate degree	35 (27.8)	5.14 (0.76)	
Bachelor’s degree or above	15 (11.9)	5.47 (0.45)	
Work status			
Retirement	66 (52.4)	5.21 (0.62)	0.062
Employment	40 (31.7)	5.18 (0.76)	
Others	20 (15.9)	4.8 (0.76)	
Monthly income (RMB)			
≤1999 (≤USD 309.2)	20 (15.9)	5.00 (0.70)	0.579
2000–3999 (USD 309.2–USD 618.6)	50 (39.7)	5.17 (0.72)	
4000–5999 (USD 618.6–USD 928.0)	36 (28.6)	5.12 (0.63)	
6000–7999 (USD 928.0–USD 1237.4)	14 (11.1)	5.14 (0.83)	
≥8000 (≥USD 1237.4)	6 (4.7)	5.54 (0.71)	
Inpatient setting			
Designated hospital isolation ward	100 (79.3)	5.26 (0.62)	0.002
Mobile cabin hospital isolation ward	26 (20.7)	4.69 (0.82)	
Inpatient days in isolation wards			
≤10	14 (11.1)	5.31 (0.55)	0.054
11–20	32 (25.4)	4.87 (0.89)	
21–30	32 (25.4)	5.12 (0.76)	
≥31	48 (38.1)	5.28 (0.49)	
Communication mode			
Face to face	119 (94.4)	5.18 (0.67)	0.004
Others	7 (5.6)	4.40 (0.91)	
Previous hospitalization experiences			
Yes	23 (18.3)	5.17 (0.69)	0.811
No	103 (81.7)	5.13 (0.71)	

**Table 3 ijerph-20-02717-t003:** Comparison of nurse and patient self-evaluated CBI-24 scores.

CBI-24 Subscales	Nurses	Patients	Range	*p*
	Mean (*SD*)	Mean (*SD*)		
Assurance of human presence	5.38 (0.78)	5.27 (0.73)	1–6	0.174
Professional knowledge and skills	5.39 (0.74)	5.35 (0.71)	1–6	0.614
Patient respectfulness	5.30 (0.73)	5.01 (0.79)	1–6	0.001
Positive connectedness	5.17 (0.81)	4.87 (0.86)	1–6	0.001
Total CBI-24 score	5.32 (0.72)	5.14 (0.70)	1–6	0.023

Abbreviations: SD—Standard Deviation; CBI-24—Caring Behaviors Inventory-24.

**Table 4 ijerph-20-02717-t004:** Nurse and patient self-evaluated CBI-24 items.

CBI-24 Items	Nurses	Patients
	Mean (*SD*)	Mean (*SD*)
#1. Attentively listening to the patient	5.37 (0.96)	5.07 (0.89)
#2. Giving instructions or teaching the patient	5.29 (0.93)	4.94 (0.95)
#3. Treating the patient as an individual	5.06 (1.04)	4.63 (1.19)
#4. Spending time with the patient	4.94 (1.06)	4.29 (1.37)
#5. Supporting the patient	5.43 (0.79)	5.13 (0.92)
#6. Being empathetic or identifying with the patient	5.43 (0.87)	5.17 (0.87)
#7. Helping the patient grow	5.24 (1.04)	5.02 (0.99)
#8. Being patient or tireless with the patient	5.39 (0.96)	5.37 (0.80)
#9. Knowing how to give shots, IVs, etc.	5.42 (0.99)	4.99 (1.37)
#10. Being confident with the patient	5.40 (0.95)	5.41 (0.83)
#11. Demonstrating professional knowledge and skill	5.26 (0.93)	5.49 (0.72)
#12. Managing equipment skillfully	5.26 (0.96)	5.46 (0.79)
#13. Allowing the patient to express feelings about his or her disease and treatment	5.2 3(1.00)	4.98 (1.10)
#14. Including the patient in planning his or her care	4.98 (1.05)	4.75 (1.15)
#15. Treating patient information confidentially	5.61 (0.78)	5.40 (0.86)
#16. Returning to the patient voluntarily	5.44 (1.01)	5.18 (1.01)
#17. Talking with the patient	5.30 (0.97)	4.93 (1.10)
#18. Encouraging the patient to call if there are problems	5.51 (0.87)	5.37 (0.89)
#19. Meeting the patient’s stated and unstated needs	5.26 (0.92)	5.10 (1.00)
#20. Responding quickly to the patient’s calls	5.41 (0.97)	5.45 (0.79)
#21. Helping to reduce the patient’s pain	5.23 (0.93)	5.10 (1.05)
#22. Showing concern for the patient	5.48 (0.89)	5.33 (0.89)
#23. Giving the patient’s treatments and medications on time	5.47 (0.95)	5.60 (0.66)
#24. Relieving the patient’s symptoms	5.20 (1.03)	5.15 (0.95)

Abbreviations: CBI-24—Caring Behaviors Inventory-24.

**Table 5 ijerph-20-02717-t005:** Multiple linear regression analysis of the influencing factors of caring behaviors assessed by nurses and patients.

	Dependent Variable	Independent Variable	*β*	95% CI (Lower, Upper)	*p*	Adjusted R^2^
Nurses	Total CBI-24 score	Sex	0.266	(0.051, 0.481)	0.015	0.220
		Ethical training experience	0.286	(0.100, 0.471)	0.003	
		Professional title	−0.256	(−0.392, −0.120)	<0.001	
		The length of working experience in years	−0.115	(−0.218, −0.011)	0.031	
		Whether being dispatched to Wuhan	0.384	(0.100, 0.668)	0.008	
	Assurance of human presence	Sex	0.332	(0.101, 0.563)	0.005	0.234
		Ethical training experience	0.323	(0.124, 0.523)	0.002	
		Educational background	0.211	(0.044, 0.378)	0.013	
		Professional title	−0.297	(−0.442, −0.152)	<0.001	
		The length of working experience in years	−0.128	(−0.239, −0.017)	0.024	
	Professional knowledge and skills	Sex	0.285	(0.057, 0.513)	0.015	0.170
		Ethical training experience	0.269	(0.072, 0.466)	0.008	
		Professional title	−0.214	(−0.357, −0.071)	0.004	
	Patient respectfulness	Ethical training experience	0.293	(0.100, 0.486)	0.003	0.183
		Professional title	−0.271	(−0.412, −0.129)	<0.001	
		The length of working experience in years	−0.111	(−0.219, −0.002)	0.045	
		Whether being dispatched to Wuhan	0.396	(0.100, 0.692)	0.009	
	Positive connectedness	Ethical training experience	0.249	(0.031, 0.467)	0.025	0.152
		Professional title	−0.19	(−0.350, −0.031)	0.020	
		The length of working experience in years	−0.134	(−0.257, −0.012)	0.031	
		Whether being dispatched to Wuhan	0.591	(0.257, 0.925)	0.001	
Patients	Total CBI-24 score	Inpatient setting	0.516	(0.230, 0.802)	0.001	0.140
		Communication mode	0.661	(0.155, 1.166)	0.011	
	Assurance of human presence	Inpatient setting	0.605	(0.305, 0.906)	<0.001	0.106
	Professional knowledge and skills	Inpatient setting	0.670	(0.373, 0.968)	<0.001	0.134
	Patient respectfulness	Inpatient setting	0.372	(0.040, 0.704)	0.029	0.082
		Communication mode	0.759	(0.172, 1.347)	0.012	
	Positive connectedness	Inpatient setting	0.494	(0.138, 0.850)	0.007	0.106
		Communication mode	0.838	(0.210, 1.467)	0.009	

Note: *β*—partial regression coefficient.

## Data Availability

The data that support the findings of this study will be available from the corresponding author upon reasonable request.

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
