# Peer review of "Nurse and Patient Assessments of COVID-19 Care Quality in China: A Comparative Survey Study"

_ijerph, 2023, doi:10.3390/ijerph20032717_

Round 1
Reviewer 1 Report (Previous Reviewer 3)
Dear authors!
Thank you for giving me the possibility to review your manuscript regarding the “ Nurse and patient assessments of COVID-19 care quality in China: A comparative survey study”. This is a very interesting topic.
In general, the paper is very well done and written and was improved through the revision. However, I have some comments and questions, which were not solved yet.

Author Response
Dear reviewer,
We really appreciate your valuable comments on our manuscript. We have addressed these comments one by one as follows. We have highlighted the amendments by using tracking in the main text. Please see the attachment.
Thank you very much for your review and consideration.

Reviewer 2 Report (Previous Reviewer 1)
Dear Authors,
Thank you very much for the opportunity to review this manuscript again. I consider that you responde to the suggestions I propose in the first revision and you improve your manuscript respondind to the suggestions of all the reviewers and editor.
Author Response
Dear reviewer,
We really appreciate your valuable comments on our manuscript.
Thank you very much for your review and consideration.
This manuscript is a resubmission of an earlier submission. The following is a list of the peer review reports and author responses from that submission.
Round 1
Reviewer 1 Report
Dear authors,
I would like to thank you for the opportunity I have been given to review this paper. This is a well written an interesting article related with the quality-of-care perceptions of nurses and patients during the Covid 19 pandemic. There are some revisions that I suggest:
1-In page 5 please verify N for female. In the text you mention 71,9%, in the table 1 you mention 81,7%.
2- Also in this page please explain what you mean with ethical training experience. How does a nurse get this experience? Also, what´s the difference between primary nurse and senior nurse?
3- In table 4, #15 please correct the word confidentially.
4- Page 11, line 283, you mean First or Frist?
Author Response
Dear Reviewer 1,
We really appreciate your valuable comments on our manuscript. We have addressed these comments one by one as follows. We have highlighted the amendments by using tracking in the main text. In addition, we asked a languish editing company (Scribendi Inc.) to edit the grammar errors and check the languish usage throughout the whole text (including reference format checking) after we revised the content.
Thank you very much for your review and consideration.
Sincerely,
Xianhong Li

Reviewer 2 Report
Authors,
Thank you for the opportunity to review your article, “Do nurses assess the quality of care consistently with COVID-19 patients? A paralleled cross-sectional study in China.”
I believe that your work here is fatally flawed based on your findings and reporting.
Please consider the following areas of improvement:
Lines:
11-12 “exceptional caring circumstances” can be interpreted as either nurses giving exceptional care or problems with the care being given. This needs to be clarified.
15 correct punctuation in date
20 “working years” is unclear. Are you stating the length of time the individual was a nurse or the length of time at their employer? This needs to be clarified.
34-35 Your paper is about COVID-19, yet you cite a 2013 paper to draw a conclusion about it. You have—maybe—an inference here regarding what was needed. You need to find better and more current information to make this claim.
36-43 You have present and past tense sentences mixed together in the same paragraph.
54-57 Grammar.
58-59 Sentence needs clarification. “…it is expected to find…” What is?
66-69 If you did a systematic search, there needs to be further information on it, as there are many other articles on the topic than just twelve you state you found. Did you do a systematic search? If so, what protocol was used? Or did you simply just search keywords ineffectually to only find the small number you did?
71 Review the use of the word “discrepantly”
80-82 A PubMed search of ("caring behavior" AND pandemic) revealed THREE papers on this topic. Your assertions are incorrect and unresearched.
82-84 Incorrect sentence with these findings above from 80-82
92 Correct date punctuation
92-93 The correct terminology for STROBE is “Strengthening the reporting of observational studies in epidemiology.” Additionally, you have no citation or reference for the version of STROBE which you used. I do not see how this paper is epidemiologically based, nor how STROBE protocols are apropos to your work.
100-102 Review parallel sentence construction.
110-111 “Nurses who were suspected or confirmed with 110 COVID-19 infection were excluded.” Why? You missed an entire population of nurses, excluding them. Grammar as well.
117 “Suspected patients and severely ill patients were excluded from the study.” Why would you exclude severely ill patients from your study?
122-126 “whether being dispatched to Wuhan” What does this mean?? If they were working at the hospitals from which you derived your convenience sample, they were working there.
140-142 You state a Cronbach’s of .96/.97—but the study you cite was not the tested original. Are you making assumptions from one study [19] and attempting to apply them to yours? This is not statistically sound.
145 Correct date punctuation.
159 ” No personal identifiers were collected” This is incorrect, as much of the information you collected as “demographic information” could easily be put together to figure out who the respondents were. You collected personally identifiable information.
162 “The study was conducted according to general ethical standards” Whose “general ethical standards?”
163 “approved by the Institutional Review Board of XXX” As XXX is nobody, am I to assume that you did not follow proper IRB standards?
184 “and 90.6% were dispatched to Wuhan” If you were looking specifically at nurses at Wuhan hospitals and surveyed only specific Wuhan hospitals, I do not understand why this number is not 100%
192 Table 2 has significant formatting areas under “variables”
204 Table 3 has odd formatting under subscales with bold and underlining
207 “patienarding” Was this paper proofread? What does this even mean??
212 Table 4 also has odd formatting with bold
221 having is the incorrect tense for the verb
221-222 Again if the nurses were not dispatched to Wuhan, why are they included in this study?
236-237 Again, there are three, as discussed earlier, all surrounding COVID-19.
244-247 I am uncertain how you can make this assumption based on your data or the data you cite in [9].
247-249 Several areas of the paper state that there were no studies known which looked at this topic, yet this sentence exists “Moreover, the scores of caring behaviors in our study were much higher 247 than those reported in other studies both from nurses’ and patients’ perspectives 248 [19,21,26,30,33].”
249-253 Conclusions are not based in fact or on the study completed. Reference [37] is an unscholarly newspaper article.
272-273 “traditional “patient-272 centered” care was shifted to “public health centered” care (to prevent occupational infection) [43]” What does this even mean? This was not discussed anywhere in your paper at all.
DISCUSSION, in general. There is not enough factual information here to make most of the conclusions that the authors are attempting to make. Most of this portion of the paper is based on conjecture.
316-319 Assumptions are made without validation or support regarding experienced nurses perceptions and expectations of care delivery.
345-351 Nearly twelve pages of information and one paragraph of conclusions from your paper? Was this paper worth investigating to begin with?
363 “Informed consent statement” Not applicable?????
REFERENCES formatting is inconsistent between individual reference sources.
Author Response
Dear Reviewer 2,
We really appreciate your valuable comments on our manuscript. We have addressed these comments one by one as follows. We have highlighted the amendments by using tracking in the main text. In addition, we asked a languish editing company (Scribendi Inc.) to edit the grammar errors and check the languish usage throughout the whole text (including reference format checking) after we revised the content.
Thank you very much for your review and consideration.
Sincerely,
Xianhong Li

Reviewer 3 Report
The specific comments are shown in the attached file.

Author Response
Dear Reviewer 3,
We really appreciate your valuable comments on our manuscript. We have addressed these comments one by one as follows. We have highlighted the amendments by using tracking in the main text. In addition, we asked a languish editing company (Scribendi Inc.) to edit the grammar errors and check the languish usage throughout the whole text (including reference format checking) after we revised the content.
Thank you very much for your review and consideration.
Sincerely,
Xianhong Li

Round 2
Reviewer 2 Report
You extensive improvements through dedicated hard work have significantly improved this paper. Congratulations on your publication.